# Pre-Saturation of Bran as a Strategy for Developing Oat Bran-Enriched Bread

**DOI:** 10.3390/foods14122071

**Published:** 2025-06-12

**Authors:** Yun Wu, Tao Wang, Maria Ortiz de Erive, Guibing Chen

**Affiliations:** 1Center for Excellence in Post-Harvest Technologies, North Carolina A&T State University, The North Carolina Research Campus, 500 Laureate Way, Kannapolis, NC 28081, USA; ywu@aggies.ncat.edu (Y.W.); twang@ncat.edu (T.W.); mortizde@ncat.edu (M.O.d.E.); 2Department of Energy and Environmental Systems, North Carolina A&T State University, 1601 E Market Street, Greensboro, NC 27411, USA

**Keywords:** oat bran, bread, pre-saturation, water competition, gluten network, water activity

## Abstract

Oat bran offers notable health benefits, but excessive incorporation into bread often compromises quality and consumer acceptance due to its competition for water, particularly with gluten, impairing dough structure. The pre-hydration of fibrous ingredients could alleviate their negative impact on bread quality. This study aimed to determine the optimal pre-hydration level of oat bran to achieve maximal quality in bread enriched with pre-hydrated oat bran that replaced 20% white flour in a white bread formula. Oat bran was pre-hydrated to six water activity (*a*_w_) levels, ranging from 0.9951 to 0.9989. The results revealed that oat bran hydrated near its saturation point (*a*_w_ = 0.9979) yielded the composite bread with the most desirable structural and textural properties, including the highest specific loaf volume, minimal crumb hardness, and superior springiness and cohesiveness—attributes comparable to those of the control white bread. Conversely, hydration levels either below or above this saturation threshold led to a decline in bread quality. Optimally saturated oat bran exhibited significantly reduced water absorbency, ensuring sufficient water availability for gluten network development. The findings underscore the critical role of precise hydration in optimizing the functional properties of oat bran for bread-making applications.

## 1. Introduction

Oat bran, derived from the outer layers of the oat kernel during the milling process, is a nutritionally dense by-product rich in dietary fiber, particularly soluble fiber in the form of beta-glucan, which accounts for no less than 5.5% of its dry weight. It is also a significant source of protein (approximately 24%), essential unsaturated fatty acids, and a broad spectrum of micronutrients, including B vitamins (notably thiamine and folate), iron, magnesium, phosphorus, and zinc [1,2]. Furthermore, oat bran contains unique bioactive phytochemicals like phenolic acids, flavonoids, and avenanthramides—exclusive to oats—which possess notable antioxidant properties [3,4]. Oat bran’s distinct nutritional and phytochemical profile underscores its role as a functional food with multiple health-promoting effects. Incorporating oat bran into bread formulations at higher inclusion rates represents a nutritionally promising strategy to boost dietary fiber intake, potentially leading to beneficial effects in the areas of lipid metabolism, glycemic regulation, oxidative stress mitigation, and overall cardiovascular and metabolic health [5].

However, incorporating cereal bran into bread formulations poses significant technological challenges since it can negatively impact dough rheology, bread quality, and sensory acceptability. Notable adverse effects include a denser and firmer crumb structure, a coarse and gritty mouthfeel, a darkened crumb color, and undesirable bitter or off-flavors, which can deter consumer acceptance [6,7,8]. Several mechanisms underlie these detrimental effects. Physically, bran particles interfere with the formation of gluten network by serving as inert barriers, leading to both the dilution and structural weakening of the starch–gluten matrix [9]. Chemically, ferulic acid (FA), a predominant hydroxycinnamic acid in oat bran, can promote cross-linking reactions either among polysaccharide molecules or between polysaccharides and proteins. These interactions significantly affect the aggregation behavior of gluten [10,11]. In the presence of insoluble dietary fiber (IDF), FA promotes the formation of a compact IDF–FA–gluten complex, leading to a denser crumb structure [10,12,13]. Additionally, the high water-binding capacity of bran restricts water availability for gluten hydration during mixing and starch gelatinization during baking. This leads to insufficient starch swelling, the partial dehydration of the gluten matrix, and subsequent conformational alterations, ultimately destabilizing the gluten polymeric network [14,15]. A recent study suggests that the detrimental effects of bran on bread quality are governed by the interplay between water binding, primarily attributed to the insoluble bran fraction, and the plasticizing properties of the soluble bran fraction [16]. Moreover, the soluble bran fraction, notably beta-glucan, increases the viscosity of the aqueous phase within the dough and modifies the plasticizing properties of the gluten–starch matrix, subsequently impairing the dough’s rheological properties and the final crumb texture [17,18]. A more recent study demonstrated that beta-glucan weakened the quality of gluten proteins mainly through two mechanisms; one was that beta-glucan, when mixed with water, formed a highly viscous gel that attached to the surface of the gluten network structure, resulting in a discontinuous gluten network, and the other was that beta-glucan influenced the secondary and primary structures of gluten and thus destroyed the stability of the gluten network structure [19].

Various bran pretreatment techniques, such as fractionation, electro-separation, extrusion, and enzymatic modification, have been employed either individually or in combination to counteract the detrimental effects of bran in food applications [20,21,22]. Other approaches involve supplementing bread formulations with vital wheat gluten or dough strengtheners including surfactants, enzymes, and hydrocolloids [23]. Despite their effectiveness, these methods often lead to higher production expenses and may involve chemical additives, which may discourage consumers seeking products with natural ingredients. A recent study conducted in our laboratory demonstrated that high-quality bread could be successfully produced with up to 22% microfluidized corn bran by simply adjusting the dough hydration through the addition of extra water [24].

Water is a fundamental ingredient in bread-making, serving multiple vital functions in dough development and the final quality of baked bread. It enables the dissolution and hydration of dry ingredients, activates yeast metabolism, promotes gluten network development, and supports the establishment of the dough’s supramolecular architecture. Moreover, during the mixing process, water creates a continuous ‘liquid phase’ that encapsulates air bubbles, which subsequently grow and expand as fermentation progresses [25,26]. Previous research has shown that even minor variations in water content can significantly impact bread quality [24]. Both insufficient and excessive water levels in bread dough adversely affect loaf volume by disrupting gluten formation and gas retention. Insufficient hydration impedes the development of a robust gluten network, compromising gas retention and yielding a denser, smaller loaf. Conversely, excessive water dilutes the proteins and restricts their ability to interact and form a strong network. Consequently, the dough becomes overly sticky and lacks the necessary structure to support the rising gas, resulting in a collapsed or flattened loaf. In a previous study, a water content of 41.9% was identified as optimal for composite bread in which 20% of the white flour was substituted with microfluidized corn bran [24]. Other approaches, such as pre-hydrating or soaking bran, have also been explored [27,28,29]. However, these studies frequently depend on empirical trial-and-error approaches to identify optimal water levels, which can be labor-intensive and require repeated experimentation when the bran substitution level is altered.

To address these limitations, this study aimed to develop high-quality oat bran-enriched bread through a new strategy to effectively counteract the water competition effects through the pre-saturation of oat bran. The effectiveness of this novel strategy was assessed based on key quality attributes of bread, including the specific loaf volume, pore size distribution, crumb microstructure, and textural characteristics. The information generated by this study is used to provide an innovative methodology for rapid and precise estimations of the optimal hydration levels for diverse high-fiber ingredients.

## 2. Materials and Methods

### 2.1. Materials

Oat bran with 9.1% moisture, 6.2% fat, and 17.5% total dietary fiber was obtained from Bob’s Red Mill (Milwaukie, OR, USA). The bran was ground to a mean particle size of 159.9 ± 3.7 μm using an ultra-centrifugal mill (Retsch ZM200, Haan, Germany). Unbleached white all-purpose flour containing 73.5% total carbohydrate, 11–12.5% protein, 1.5% total fat, and 3% dietary fiber was acquired from Bob’s Red Mill (Milwaukie, OR, USA). Other bread-making ingredients including Fleischmann’s active dry yeast, granulated sugar, table salt, Nestle Carnation instant nonfat dry milk, and Crisco all-vegetable shortening were obtained commercially from a local supermarket.

### 2.2. Pre-Hydration Procedure of Oat Bran and Water Activity Measurement

To assess the effect of pre-hydration levels of oat bran on bread baking performances, bran samples were hydrated with varying amounts of deionized water to achieve water contents of 44.4%, 54.55%, 56.52%, 60.00%, 61.54%, 62.96%, 64.29%, and 65.52%. These values correspond to water-to-dry bran ratios of 0.8, 1.2, 1.3, 1.5, 1.6, 1.7, 1.8, and 1.9, respectively. The selected ratios were based on preliminary experiments investigating the relationship between water activity (*a*_w_) and water content, with the aim of identifying the saturation point of oat bran. Following hydration, the samples were sealed and incubated for 2 h to reach equilibrium. The aw value of the hydrated bran was then measured using an Aqualab 4T water activity meter (Decagon Devices, Pullman, WA, USA). The water content at which bran became saturated was determined from the resulting *a*_w_–water content curve.

### 2.3. Measurements of Flour and Dough Properties

#### 2.3.1. Farinograph Tests

The water absorption properties of white and composite flours (a mixture of bran and white flour) were determined using a Brabender Farinograph (C.W. Brabender Instruments Inc., South Hackensack, NJ, USA) in accordance with AACC Method 54-21A [30], employing a 50 g mixing bowl.

#### 2.3.2. Dough Extensibility

Dough was prepared using a home bread maker (Model BB-SSC10, Zojirushi Corp., Osaka, Japan). The ingredients, including white or composite flour, water, and other components outlined in Section 2.4 of the bread formulation, were subjected to an initial kneading for 12 min, followed by a 10 min resting period. A second kneading phase was then carried out for an additional 10 min. Dough extensibility was assessed using the Kieffer dough and gluten extensibility rig mounted on a TA-XT2i texture analyzer (Stable Micro Systems Ltd., Surrey, UK), in accordance with the AACC Method 54-10.01 [31]. Each dough sample was tested in ten replicates, and the mean values were recorded.

### 2.4. Bread-Making Procedure

The white bread formulation, based on 100 g white flour (14% moisture basis), consisted of 9 g sugar, 4.2 g dry milk powder, 1.5 g salt, 1.5 g yeast, and 6 g shortening. Water was added to achieve a dough consistency of 500 Brabender Units (BU), as determined by Farinograph analysis. When a portion of the white flour was substituted with oat bran (14% moisture content), the amount of water corresponding to the replaced flour portion was proportionally reduced, while the bran was pre-hydrated to varying extents before being incorporated into the standard bread formula. Bread was prepared using a home bread maker (Model BB-SSC10, Zojirushi Corp., Osaka, Japan), following the manufacturer’s standard program as outlined in our previous study [24]. Upon the completion of baking, the loaves were removed from the pans, allowed to cool to room temperature, and sealed in polyethylene bags for subsequent analysis.

### 2.5. Bread Quality Evaluation

#### 2.5.1. Specific Loaf Volume

The volume of the loaves was determined using a benchtop laser-based Volscan Profiler (Stable Micro Systems, Godalming, UK). The specific loaf volume was calculated according to Equation (1).(1)Specific loaf volumecm3g=Loaf volume (cm3)Bread weight (g)

#### 2.5.2. Texture Profile Analysis

A TA-XT2i texture analyzer (Stable Micro Systems Ltd., Surrey, UK) equipped with the “Texture Expert” software (Version 6, 1, 27, 0) was employed to evaluate the textural properties of bread crumb, specifically hardness, springiness, and cohesiveness, at ambient temperature (24 °C) according to a previously established method [32].

The experimental procedure was as follows: Two bread slices, each with a standardized thickness of 25 mm, were taken from the central portion of the loaf. A Texture Profile Analysis (TPA) test was performed using a double-compression method, wherein the samples were compressed to 50% of their original height at a constant speed of 2 mm/s. A cylindrical probe (25 mm diameter) was applied to the top, middle, and bottom regions of each slice, with a 30 s interval between the first and second compressions. The mean values for each textural attribute were subsequently calculated.

#### 2.5.3. Microstructure Analysis

The microstructure of white bread and three representative bran-enriched bread crumb samples was examined using a Zeiss EVO LS environmental scanning electron microscope (ESEM) (Carl Zeiss Microscopy Ltd., Cambridge, UK), following a previously established methodology [33]. The crumb specimens were mounted onto a sample holder and imaged at an accelerating voltage of 20 kV with 60× magnification. Additionally, higher-resolution imaging was performed at 10 kV with magnifications of 350× and 1000× to examine the detailed microstructure of the crumb matrix.

#### 2.5.4. Porosity

A bread specimen (70 mm × 4 mm × 25 mm) was cut from the central portion of a loaf and subsequently weighed. The true volume of the sample was determined using an AccuPyc II 1340 Automatic Gas Pycnometer (Micromeritics Instrument Corp., Norcross, GA, USA), while the total volume was measured by a benchtop laser-based Volscan Profiler (Stable Micro Systems, Godalming, UK). The porosity of the bread crumb was calculated according to Equation (2):(2)Porosity of crumbmm3mm3=Total volumemm3−True volume (mm3)Total volume (mm3)

#### 2.5.5. Image Analysis for Crumb Grain Characteristics

The pore size distribution within the bread crumb was analyzed by scanning the bread slices with a Bizhub C224 flatbed scanner (Konica Minolta, Oldsmar, FL, USA), which provided high-resolution digital images in real time. The acquired images were subsequently analyzed using the particle analysis tool in ImageJ^®^ software (Version 1.8.0) [34] to quantify the pore area and total pore count.

### 2.6. Statistical Analysis

Analysis of variance (ANOVA) was conducted using SPSS (Version 19. SPSS Inc., Chicago, IL, USA) statistical software to evaluate the influence of flour replacement and water content levels on the quality attributes of composite bread. Significant differences among samples were assessed using Duncan’s test at a 95% confidence level (*p* < 0.05). All analyses were performed in duplicate, and the results were expressed as mean ± standard deviation (SD).

## 3. Results and Discussion

### 3.1. The Relationship Between A_w_ and Water Content in Pre-Hydrated Oat Bran

Water activity (*a*_w_) is a thermodynamic parameter defined as the ratio of the equilibrium vapor pressure of water within a given material to that of pure water at the same temperature. So, *a*_w_ in pure water is exactly 1.0000. Water molecules in a food matrix are bound through various interactions with food components like proteins, carbohydrates, and minerals, primarily through hydrogen bonding and hydrophobic interactions. These interactions alter the properties of the water, making it less free and available than in pure water. As a result, the maximum *a*_w_ in a food matrix can only asymptotically approach, but never exactly reach, 1.0000, even as the water content increases towards a saturated state [35].

The *a*_w_ values of pre-hydrated oat bran samples as a function of water content are presented in Figure 1. A pronounced increase in *a*_w_ was observed as the water content rose from 44% to 61.54%, Beyond this critical hydration point (61.54%, *a*_w_ = 0.9979), further increases in water content resulted in only marginal rises in *a*_w_, suggesting that the bran had reached its saturation threshold. Additionally, the presence of free water was observable in the samples beyond this point, further confirming that the bran had reached its maximum absorption capacity. In experimental design, preliminary tests were conducted to narrow down the range of water content where the saturation point would be reached. Subsequently, tests with small increases in water content were conducted to identify the very narrow range of water content where the saturation point landed, which was taken as the saturated water content. Apparently, a denser sampling of water content or water activity levels could result in a more precise determination of the saturation point.

According to thermodynamic principles, mass transfer is governed by gradients in chemical potential, whereby water migrates from regions of higher chemical potential to those of lower chemical potential [36]. The chemical potential (µ) of water depends on both temperature and *a*_w_.

Within the hygroscopic range (i.e., *a*_w_ < 1), µ can be determined using Equation (3) [37](3)$$\mu =\mu_{0}+RT\ln(a_{w})$$ where $\mu_{0}$ (J/mol) is the chemical potential of pure water at temperature *T* (°K) and *R* is the ideal gas constant 8.31472 J mol^−1^ K^−1^.

According to Equation (3), the chemical potential is solely dependent on *a*_w_ under constant temperature conditions. Consequently, as pre-hydrated bran approaches a fully saturated state—where its *a*_w_ nears 1.0000—the chemical potential of water within the bran reaches its maximum value. Under such conditions, the saturated bran no longer competes with proteins (particularly gluten) for water during the mixing of composite dough. As a result, the adverse effects of untreated oat bran on the quality and sensory attributes of the final bread product are significantly reduced.

### 3.2. Impact of Oat Bran Pre-Hydration Levels on Flour and Dough Characteristics

The incorporation of oat bran into bread formulations adversely impacts the textural and sensory characteristics of the final product. This deterioration is attributed to multiple interrelated mechanisms, including the dilution of gluten proteins, interference with gluten network development, water redistribution and competition, physical hindrance by bran particles, and the release of beta-glucans from bran. A critical substitution threshold exists, beyond which the sensory quality of the bread becomes unacceptable [7,23,24]. In preliminary trials, composite flours were prepared with varying oat bran inclusion levels (10%, 15%, 20%, 25%, and 30%, *w*/*w*) to determine the maximum substitution level. For each formulation, different water content (1.2%, 2.4%, 3.6%, and 4.8% above the water level used in the white bread control) were tested to identify the optimal hydration for maximizing the specific loaf volume. The results revealed a pronounced decline in specific loaf volume when the substitution level exceeded 20%, aligning with previous findings [24,38]. Consequently, a composite flour containing 20% oat bran was selected for subsequent investigation.

When oat bran is incorporated into bread, oat beta-glucan released from it can weaken the gluten network [19]. However, the process results in a relatively minor release of this fiber, because it is primarily found in the cell walls of the oat bran, which can be resistant to solubilization, especially under the condition of limited free water in dough. Moreover, in the small range of *a*_w_ (0.9951 to 0.9989) in oat bran examined in the present study, the amount of beta-glucan released in each case could be considered the same. So, its effects on dough and bread properties could be treated as a constant when examining the impact of the bran’s capacity to absorb water when pre-hydrated to different *a*_w_.

#### 3.2.1. Farinograph Analysis

Table 1 summarizes the impact of oat bran pre-hydration on farinograph parameters. Oat bran incorporation reduced dough consistency compared to control white dough. This decline was primarily due to gluten network weakening, as oat bran diluted gluten, competed for water, and disrupted the dough matrix. The high fiber content, particularly beta-glucan, played a central role by absorbing substantial amounts of water [16], thereby limiting gluten hydration and reducing the dough’s elasticity. Bran particles also physically hindered gluten formation, yielding a weaker dough. Additionally, oat bran reduced the overall starch content in the dough and impeded starch granule hydration and swelling, which further contributed to the decrease in dough viscosity and consistency [39,40]. Consequently, higher bran substitution levels produced less elastic, fragile dough, resulting in denser bread with a reduced volume and inferior texture [41]. To counteract these adverse effects, optimizing hydration and processing conditions is essential.

In industrial baking practices, white bread dough typically requires 500 Brabender Units (BU) for optimal proofing performance and desirable bread texture and flavor. However, Table 1 reveals that the consistency of composite dough fell below this threshold and dropped from 526 BU to 237 BU as the *a*_w_ of the pre-hydrated bran increased from 0.9951 to 0.9989. Higher aw in pre-hydrated bran indicated a reduced water uptake by the bran, thereby leaving more free water available for other dough constituents. While this potentially mitigated the bran’s detrimental impact on the gluten network and positively influenced dough consistency, it also increased the hydration of other dough components such as starch, which led to a decrease in dough consistency. The net effect was a decrease in dough consistency, suggesting that the latter factor was more dominant as pre-hydration levels increased. Further analysis showed that composite dough with low pre-hydration bran was hard and stiff due to insufficient free water for gluten development, impairing gas retention and proofing. As the pre-hydration level increased, less water was absorbed by the bran, allowing more free water to lubricate the dough matrix and improve its structure [42]. Nevertheless, excessive water can also be detrimental, causing overexpansion during baking and subsequent loaf collapse. In addition, this over-dilution weakened the gluten matrix, resulting in a weaker crumb structure with low extensibility and compression resistance and ultimately a lower specific loaf volume [43,44]. These findings underscore the importance of identifying an optimal water absorption level for bran-enriched dough. Notably, the results also confirmed that neither traditional farinograph analysis based on a fixed 500 BU standard nor mixograph analysis adequately predicted water absorption for high-fiber doughs [24,45,46]. In this investigation, the best overall bread quality was observed at a maximum dough consistency of 366 BU. However, this benchmark may not apply to other bran types due to varying water absorption capacities. Therefore, establishing a new standardized method is imperative to optimize water content in bread formulations comprising high levels of diverse high-fiber ingredients.

The dough development time (DDT) serves as a reliable indicator of dough strength under mechanical stress [47]. As indicated in Table 1, white dough, with a higher gluten content and efficient water absorption, exhibited a shorter DDT than composite dough. In the absence of bran, water was rapidly and uniformly distributed among flour components, facilitating the development of a robust gluten protein network. Furthermore, the lack of bran eliminated potential structural disruptions, enabling unimpeded gluten development. As a result, white dough achieved an optimal consistency more rapidly than composite dough. Notably, the data revealed that increasing the *a*_w_ of pre-hydrated bran significantly increased the DDT of the composite dough, indicating that a longer time was needed for adequate dough formation. This is likely due to the reduced competition for water in highly hydrated bran, which made more water available for gluten network formation and consequently required prolonged mixing times [48]. Previous studies have demonstrated that the DDT of composite dough is influenced by the bran type, substitution level, and pre-hydration degree [45,46].

#### 3.2.2. Uniaxial Extensional Properties of Dough

Dough extensibility is a fundamental attribute that significantly influences the handling properties, leavening potential, and structural integrity of baked products. The dough extensibility test generates a resistance force vs. extension curve. From this curve, key quantitative parameters can be extracted to evaluate dough performance. Specifically, the R_max_ value denotes the dough’s resistance force during extension and serves as an indicator of tensile strength and gluten network robustness. The EX_max_ value, which measures the extension length at maximum resistance, reflects the dough’s elasticity and extensibility. The area under the curve at maximum resistance, A_max_, quantifies the total mechanical energy absorbed during deformation, representing the integrated effect of dough strength and flexibility. Collectively, these metrics provide a robust framework for assessing dough quality and predicting the quality of the final baked product [49,50].

Figure 2 presents the A_max_ values for composite dough formulations containing pre-hydrated oat bran at varying *a*_w_ levels. Composite doughs with low pre-hydration (low *a*_w_) bran exhibited a reduced A_max_ value, primarily due to having a very short EX_max_ despite having a relatively high R_max_, indicating a rigid and inextensible structure. These doughs were stiff and lacked extensibility, which restricted free expansion during proofing, ultimately yielding loaves with a low specific volume. In contrast, dough incorporating pre-saturated bran (*a*_w_ = 0.9979) demonstrated an A_max_ value comparable to that of white flour dough. This suggests a balanced combination of extensibility and elasticity, contributing to optimal proofing and baking performance. However, excessive water addition beyond the bran’s saturation point resulted in dough with a markedly low A_max_, indicating weakened structural integrity and suboptimal baking performance. An ideal loaf volume can only be achieved when internal gas bubbles expand without premature rupture or structural collapse, a process governed by the balance between dough resistance and extensibility [51]. In addition, a larger A_max_ value signifies that the dough can absorb more energy during stretching, reflecting both strength and extensibility—key attributes for producing voluminous, aerated bread loaves. Among the tested composite doughs (Figure 2), the dough containing pre-saturated bran exhibited a high A_max_ value, indicative of adequate extensibility and elasticity. This balance ensured sufficient gas bubble expansion while preventing premature rupture or collapse during proofing and baking, ultimately yielding high-quality bread characterized by A large specific volume and a soft, elastic, and porous crumb structure.

In summary, bran influenced the development of composite bread dough primarily in two ways: the steric hindrance and gluten dilution of the bran and its competition for water. The former was primarily determined by its particle size distribution and concentration in the dough, while the latter depended on the extent of its pre-hydration. Based on farinograph graph tests on composite flour comprising pre-hydrated oat bran and the analysis of the uniaxial extensional properties of the composite dough, the pre-saturation of the bran could minimize its adverse effects on the properties of the composite dough. However, it warrants exploration at the molecular level to determine how pre-saturated bran impacts the dynamics of water and the complex interactions between water and other components in composite bread dough during kneading and proofing. Therefore, in future studies, the low-field nuclear magnetic resonance (LF-NMR) technique should be used to characterize water mobility and distribution in the dough following a reported procedure [52].

### 3.3. Impact of Oat Bran Pre-Hydration Levels on Specific Loaf Volume

The specific loaf volume is a key indicator of bread quality. In preliminary experiment, we examined the effect of substituting white flour with an equal proportion of dry bran and pre-hydrated bran (*a*_w_ = 0.9951–0.9989) on the specific loaf volume. Even under the identical water addition levels, bread containing pre-hydrated bran exhibited a significantly higher specific loaf volume compared to bread with dry bran. Therefore, the adequate pre-hydration of oat bran appeared to be a critical step in overcoming its detrimental effects on bread quality.

As shown in Table 2, the specific loaf volume of bread formulated with pre-hydrated oat bran varied with *a*_w_ (between 0.9951 and 0.9989). The volume increased progressively as *a*_w_ rose, reaching its maximum when bran was saturated (*a*_w_ = 0.9979). Beyond this point, further water addition led to a gradual decline in specific loaf volume. These findings clearly indicate that saturated bran exerted the least detrimental impact on loaf volume, consistent with the observed improvements in dough consistency and extensibility when pre-hydrated bran was incorporated.

Both the water-binding capacity and the physical presence of bran are key determinants of composite bread quality [20,45]. Although their relative contributions remain under debate, emerging evidence indicates that bran’s hydration properties play a dominant role [16,45]. In low-hydration oat bran systems, water migration and competition between the bran matrix and gluten network induced partial dehydration and structural disruption in gluten proteins. This resulted in discontinuous and fragmented gluten networks, impairing gas retention and promoting gaseous release. Conversely, excessive water addition diluted the gluten network, reducing dough strength and viscoelasticity. This led to poor fermentation stability, a diminished gas-holding capacity, and reduced resistance to expansion during baking, collectively decreasing the specific loaf volume. Precise water optimization is therefore critical in achieving the maximum specific loaf volume [24], as it promotes the development of a robust gluten network and enables the formation of a distinct aqueous phase within the dough. This phase aids in stabilizing the air/water interface surrounding the gas cells by surface-active components, thereby enhancing gas retention and overall bread quality [53,54].

### 3.4. Impact of Oat Bran Pre-Hydration Levels on Textural Properties of Bread Crumb

The incorporation of oat bran affected the bread crumb texture, with increased hardness and reduced springiness and cohesiveness observed compared to the white bread control (Table 2). As demonstrated in reported studies, water binding by bran predominately affects dough and bread quality, while the role of steric hindrance and gluten dilution is limited [16,45]. Consequently, the adverse effects of bran addition could be alleviated by the bran’s pre-hydration level.

Bread crumb hardness showed an inverse relationship with the specific loaf volume. As shown in Table 2, the hardness decreased as the bran’s *a*_w_ or pre-hydration level increased, reaching a minimum at the pre-saturation point (*a*_w_ = 0.9979). Beyond this point, the hardness rose again. A sharp decline in hardness when *a*_w_ varied from 0.9951 to 0.9979 coincided with a significant increase in the specific loaf volume, supported by ESEM micrographs (Figure 3) showing a more uniform crumb microstructure at the pre-saturation point, which contributed to reduced hardness.

Springiness, reflecting crumb elasticity, and cohesiveness, indicating the internal bond strength, both peaked at the same pre-saturation point (*a*_w_ = 0.9979) before declining with further hydration (Table 2). Improved textural qualities were linked to its uniform, highly porous microstructure. This structure favored the crumb’s ability to recover its shape following deformation during compression testing. Previous studies have shown that an optimal water content in composite dough plays a critical role in stabilizing the dough–gas interface, thereby improving the gas retention capacity and minimizing gas cell collapse or coalescence during the fermentation and baking processes [55]. Additionally, sufficient water availability ensures the complete gelatinization of starch granules during baking. This facilitates the interactions between gluten proteins and gelatinized starch, forming an interconnected sponge-like crumb structure, thereby improving the crumb’s elasticity and ultimately influencing its mechanical properties [56]. Moreover, adequate water in bread formulations is essential in achieving optimal adhesion strength between starch granules and the gluten matrix. Prior research also found that water distribution within the gluten matrix and between starch amorphous and crystalline regions significantly impacted starch granule integrity and the adhesion strength between starch and gluten [57].

Notably, the optimal water content required to achieve desirable bread quality varied across studies, primarily due to differences in the bran type, treatment methods, and experimental conditions. For instance, a prior investigation identified 41.9% as the optimal water content for composite bread containing 20% microfluidized corn bran [24]. In contrast, using the same white bread formulation and bran replacement level, the current study determined a higher optimal water content of 45.4% when incorporating pre-saturated oat bran. These results underscore the need for empirical optimization tailored to specific bran properties.

### 3.5. Impact of Oat Bran Pre-Hydration Levels on Bread Crumb Microstructure

Figure 3 displays environmental scanning electron microscopy (ESEM) images of the crumb structure of white bread (control) and composite bread containing pre-hydrated oat bran at varying *a*_w_ values, i.e., 0.9951 (undersaturated), 0.9979 (saturated), and 0.9989 (oversaturated), corresponding to a low, optimal, and high hydration state, respectively. The control bread exhibited a continuous sponge-like network (Figure 3a). In contrast, breads incorporating either undersaturated (Figure 3b) or oversaturated (Figure 3d) oat bran displayed irregular, coarse crumb textures with non-uniform gas cell distribution and structural defects, such as cracks and large voids. Notably, the bread with optimally hydrated bran (Figure 3c) showed a microstructure comparable to the control, albeit slightly rougher due to the presence of bran particles.

Higher-magnification ESEM images (Figure 3e–l) further elucidated microstructural details. In the crumb with undersaturated bran (Figure 3j), gluten proteins formed a discontinuous matrix around starch granules, which appeared oval/spherical and ungelatinized or partially gelatinized (Figure 3f,j). The presence of aggregated starch granules and rupture zones indicated a heterogeneous gluten–starch network, attributed to insufficient hydration. For bread containing oversaturated bran, the microstructure was markedly different (Figure 3l). The gluten phase appeared to diffuse into excessively swollen starch granules, forming composite aggregates without well-defined 3D network, and starch granules became unrecognizable due to extensive gelatinization. In contrast, the crumb structure of bread made with optimally saturated bran exhibited a homogeneous, honeycomb-like porous structure, with evenly dispersed gelatinized starch granules embedded in a continuous gluten matrix (Figure 3k).

### 3.6. Grain Structure Characteristics of Bread Crumb and the Impact of Bran Pre-Hydration

The grain characteristics of bread crumb, which refer to the spatial organization and dimensions of the air cells within the crumb matrix, are critical determinants of bread quality due to their direct impact on textural and sensory attributes.

To evaluate the effects of varying pre-hydration levels of oat bran (*a*_w_ of 0.9951, 0.9979, and 0.9989, representing an undersaturated, saturated, and oversaturated hydration state, respectively) on the crumb structure of composite breads containing 20% oat bran, key parameters including the total cell count, average cell area, proportion of small cells (area < 20 mm^2^), and overall porosity were assessed. The findings are summarized in Table 3. A comparative analysis revealed that the control white bread exhibited high porosity, with large numbers for both the total and small cell counts. In contrast, all composite breads displayed reduced porosity, lower cell counts, and larger average cell sizes. Among the composite breads, the sample with saturated bran (*a*_w_ = 0.9979) displayed the most favorable crumb structure: the highest porosity, the greatest total and small cell counts, and the smallest average cell size. In comparison, the oversaturated bran sample (*a*_w_ = 0.9989) yielded the least desirable outcomes, including the lowest porosity, the lowest total and small cells, and the largest average cell size.

Water content significantly influences dough development by modulating the transmission of mechanical energy during mixing, which is essential for entrapping sufficient gas nuclei [26]. Insufficient water produces stiff dough, restricting initial gas incorporation and impeding gas cell expansion during the proofing and early baking stages [58]. Conversely, excessive water causes dough to be too soft and lacks the viscosity necessary for efficient energy transfer, impairing gluten alignment. Additionally, it may solubilize flour components that destabilize gas cells, further compromising crumb quality. An optimal hydration level promotes the formation of strong viscoelastic gluten–starch networks capable of enveloping and stabilizing individual gas cells. Such networks exhibit a strain-hardening effect during deformation, enabling uniform gas expansion and maintaining cell integrity [54]. The results of the present study underscore the importance of precise hydration in achieving optimal dough consistency and crumb structure, with properly saturated oat bran emerging as the most effective for improving composite bread quality.

## 4. Conclusions

This study clearly demonstrated that incorporating pre-hydrated but unsaturated oat bran into bread dough impaired gluten network development due to its competition for water, resulting in a weak dough structure, poor gas retention, a low loaf volume, and an undesirable bread crumb texture. In contrast, using optimally saturated oat bran could eliminate this competition, allowing normal gluten formation and yielding bread with a porous microstructure and desirable texture properties comparable to the control white bread. However, excessive water beyond the saturation of bran weakened the dough structure, leading to an inferior loaf volume and crumb quality, likely due to reduced dough strength and premature gas cell rupture. These findings indicate that the optimal quality in oat bran-enriched bread can be achieved by simply substituting a specific proportion of white flour in a standard bread formulation with pre-saturated oat bran, while proportionally reducing the water content and keeping all other non-water ingredients constant. This approach was based on the fact that when oat bran is fully saturated with water before being incorporated into a bread formula, it has no ability to compete for water with other components during bread dough development so that the gluten network in the dough can be fully developed. Theoretically, it should be broadly applicable to other high-fiber ingredients. Nevertheless, further exploration using the LF-NMR technique is needed at the molecular level into the water dynamics within fiber-enriched dough systems and the intricate interactions between water and other constituents.

## Figures and Tables

**Figure 1 foods-14-02071-f001:**
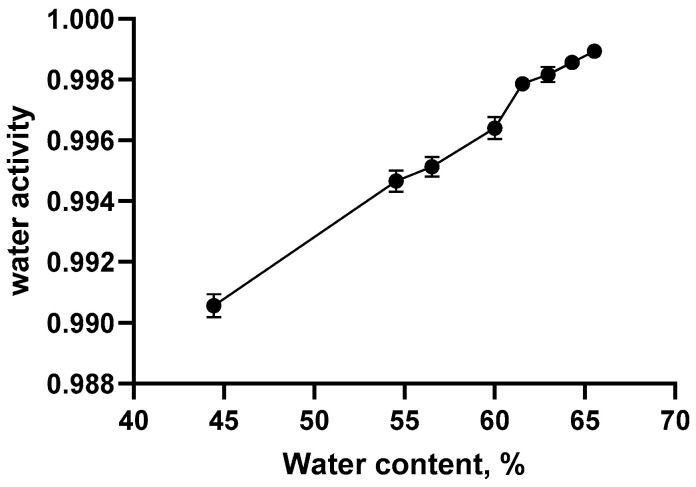
Water activity of pre-hydrated oat bran sample as a function of water content.

**Figure 2 foods-14-02071-f002:**
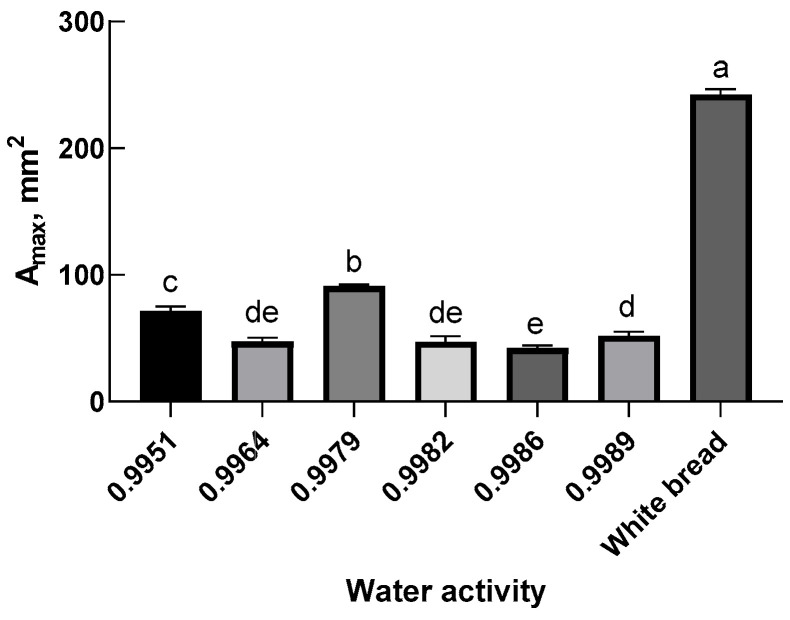
The area under the extension to the maximum resistance as a function of water activity in pre-hydrated oat bran. Different letters denote significant differences (*p* < 0.05).

**Figure 3 foods-14-02071-f003:**
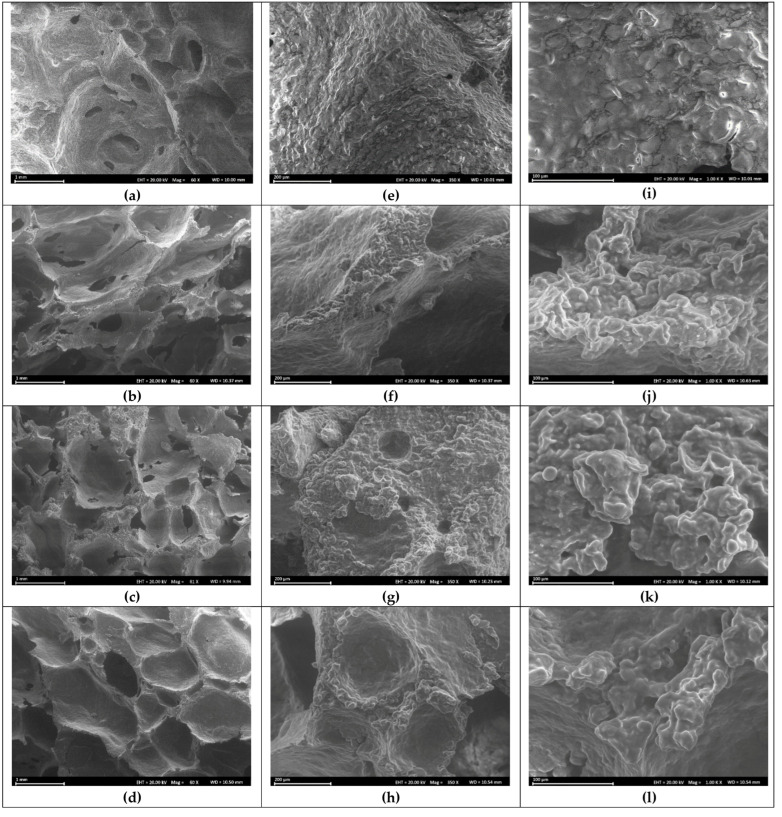
ESEM micrographs of white bread (**a**,**e**,**i**), composite bread comprising pre-hydrated oat bran with a water activity of 0.9951 (**b**,**f**,**j**), 0.9979 (**c**,**g**,**k**), and 0.9989 (**d**,**h**,**l**), respectively, (**a**–**d**) at 60× magnification; (**e**–**h**) at 350× magnification; and (**i**–**l**) at 1000× magnification.

**Table 1 foods-14-02071-t001:** Effect of *a*_w_ values of pre-hydrated oat bran with 20% flour replacement on farinograph parameters of composite dough compared to the control. In each column, different letters denote significant difference (*p* < 0.05).

Water Activity	Dough Moisture (%)	Consistency (BU)	Development Time (min)
0.9951	56.52	526 ± 12 ^a^	5.9 ± 0.2 ^a^
0.9964	60.00	435 ± 10 ^b^	10.6 ± 0.3 ^b^
0.9979	61.54	366 ± 20 ^c^	11.8 ± 0.6 ^b^
0.9982	62.96	326 ± 11 ^cd^	14.9 ± 0.4 ^c^
0.9986	64.29	316 ± 8 ^d^	17.7 ± 1.2 ^d^
0.9989	65.52	237 ± 10 ^e^	22.4 ± 1.2 ^e^

**Table 2 foods-14-02071-t002:** Specific loaf volume, porosity, and texture properties of composite bread formulated with pre-hydrated oat bran with *a*_w_ between 0.9951 and 0.9989 compared with the white bread control. In each column, different letters denote significant difference (*p* < 0.05).

Water Activity	Specific Loaf Volume (cm^3^/g)	Hardness (N)	Springiness	Cohesiveness
0.9951	3.53 ± 0.05 ^de^	554.499 ± 16.084 ^a^	0.945 ± 0.033 ^a^	0.532 ± 0.031 ^b^
0.9964	3.8 ± 0.08 ^cd^	525.748 ± 4.992 ^a^	0.952 ± 0.023 ^a^	0.582 ± 0.027 ^ab^
0.9979	4.21 ± 0.14 ^b^	346.039 ± 14.906 ^cd^	0.969 ± 0.022 ^a^	0.66 ± 0.035 ^a^
0.9982	3.85 ± 0.06 ^c^	443.633 ± 17.012 ^b^	0.963 ± 0.04 ^a^	0.621 ± 0.052 ^a^
0.9986	3.68 ± 0.11 ^cde^	409.825 ± 12.981 ^b^	0.964 ± 0.028 ^a^	0.611 ± 0.021 ^ab^
0.9989	3.47 ± 0.15 ^e^	360.43 ± 17.285 ^c^	0.951 ± 0.024 ^a^	0.59 ± 0.029 ^ab^
White bread	4.68 ± 0.06 ^a^	311.097 ± 10.653 ^d^	0.964 ± 0.007 ^a^	0.635 ± 0.01 ^a^

**Table 3 foods-14-02071-t003:** Crumb grain characteristics of white bread and composite bread comprising 20% oat bran with different hydration levels. In each column, different letters denote significant difference (*p* < 0.05).

Water Activity	Total Number of Cells	Porosity	Average Area of Cells, mm^2^	Area Range of Cells, mm^2^	Percentage of Number of Cells < 20 mm^2^, %
White	1834 ± 102 ^a^	0.9021 ± 0.012 ^a^	51.1 ± 2.2 ^a^	2632.2 ± 142 ^a^	70.4 ± 3.2 ^a^
0.9951	1464 ± 70 ^b^	0.8122 ± 0.021 ^b^	73.2 ± 2.5 ^b^	3528 ± 151.8 ^b^	63.1 ± 2.2 ^b^
0.9979	1743 ± 72 ^a^	0.8603 ± 0.0275 ^ab^	66.7 ± 2.1 ^b^	2758.6 ± 134.8 ^a^	65.7 ± 3 ^ab^
0.9989	1153 ± 41 ^c^	0.8091 ± 0.0224 ^b^	103.4 ± 5.1 ^c^	3623.1 ± 130 ^b^	59.7 ± 2.7 ^b^

## Data Availability

The original contributions presented in the study are included in the article; further inquiries can be directed to the corresponding author.

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
