# Peer review of "Pre-Saturation of Bran as a Strategy for Developing Oat Bran-Enriched Bread"

_foods, 2025, doi:10.3390/foods14122071_

Round 1

Reviewer 1 Report

Comments and Suggestions for Authors

Question 1
This paper studied the effect of oat bran prehydration to six water activity levels on the structure and quality of composite bread, including 0.9951, 0.9964, 0.9979, 0.9982, 0.9986, and 0.9989. The experimental results showed that oat bran hydrated near its saturation point (aw = 0.9979) yielded composite bread with the most desirable structural and textural properties. Since all water activities (ranging from 0.9951 to 0.9989) have not been analyzed and verified, it is impossible to prove that aw = 0.9979 is the optimal value. More comprehensive experimental data are needed to confirm this research result.

Question 2

In the process of making high-fiber bread using pre-saturated oat bran, the mechanism and principle of water activity regulation need to be clarified.

Question 3
Results and discussion should be strengthened.

Reviewer 2 Report

Comments and Suggestions for Authors

The manuscript discusses the impact of pre-hydration of oat bran at different levels on the bread baking performance. The topic of the work is interesting and should be insightful to the journal’s readership. Still, the manuscript can benefit from an inclusion of additional discussions in some sections.

I have the following comments to the authors:

  1. In lines 64-67, add a brief discussion on how beta-glucan modifies the plasticizing properties of gluten-starch matrix.

         Please see below some relevant publications:

         https://doi.org/10.1016/j.foodchem.2023.136002

         https://doi.org/10.1021/jf300786f

  1. Dough consistency is reported in Brabender units (BU). Please correct the unit to an SI unit throughout the manuscript including Figures and Tables.
  1. In lines 206-208, bring examples of the types of present interactions between solute and water molecules? Are there surface or polar interactions?
  1. In section 3.2.2, lines 304-314, “Collectively, these metrics provide a robust framework for assessing dough quality and predicting the quality of the final baked product.”

         provide an adequate reference for understanding of the reader.

  1. Referring to section 3.4, lines 375-376, add a brief discussion on the reason for the observed increase in hardness and a decrease in springiness and cohesiveness of the bread crumb texture with the incorporation of oat bran.

Reviewer 3 Report

Comments and Suggestions for Authors

For the manuscript entitled: Development of High-Fiber Bread Using Pre-Saturated Oat Bran: A Water Adjustment Strategy.

The aim of the study is to develop high-quality bread enriched with oat bran by introducing a water adjustment strategy through pre-saturation of oat bran, thereby addressing a common issue in fibre-enriched baking: the competition for water between fibre and gluten.

The main research question is: Can pre-saturating oat bran to specific hydration levels improve the structural and sensory quality of high-fibre bread by reducing water competition in dough systems? This question is both focused and practical, addressing a fundamental limitation in the formulation of fibre-rich bakery products.

The study is highly relevant in the context of growing consumer demand for high-fibre, functional foods, alongside industry efforts to develop nutritionally enhanced products without compromising on quality. It addresses a notable gap in the field: although the benefits of oat bran are well-established, its inclusion in bread formulations has traditionally resulted in undesirable texture and volume due to water competition.

Previous studies have acknowledged this problem and suggested general pre-hydration of fibrous ingredients, but few have presented a systematic, quantitative method for determining optimal hydration thresholds. The paper uniquely contributes by establishing water activity (aw) as a precise metric for managing this process.

The novelty of this research lies in several points as introducing a quantitative, reproducible method to determine the optimal water activity (aw) for pre-saturating oat bran; demonstrating for the first time that pre-hydrating oat bran to near-saturation (aw = 0.9979) enables high-fibre bread to match the quality of standard white bread; offering a practical hydration strategy that avoids additives and is easily adaptable for industry use; and proposing the generalisation of this method to other high-fibre ingredients for broader applications in functional baking.

The study bridges a critical divide between nutritional enhancement and technological feasibility in bakery product development presenting a robust and innovative approach to overcoming a longstanding challenge in fibre-enriched bread making. By establishing a quantitative, reproducible method to optimise the hydration level of oat bran, it offers a path forward for developing nutrient-dense breads with high consumer acceptability. The work’s scientific rigor, clear application potential, and contribution to the field of functional food formulation make it a valuable reference for future research and commercial development alike.

The title is long but ok.

The abstract has more than 200 words and according journal rules must be less. According to the journal's instructions the abstract presents overall main studies and used methods with the principal results and conclusions.

Keywords (presenting 6 of 10 possible) words part of the title doesn't need to be as keyword.

The introduction is well-prepared and relevant to the work.

The methodology is well described.

The first time you use an abbreviation please write it out in full firstly.

The results and discussion are extensive and well explained, usually, the results obtained are discussed and compared with those from other works so this section only needs some minor corrections.

Graphics in figure 2 doesn’t need the column borders.

Table 3 doesn’t need the inner lines and readjust the table first line to only 2 lines by last column width increasing.

Units: Please add units to variables in table 2 and also change some results considering the significant digits.

The conclusions need to be improved showing they are consistent with the evidence and arguments presented and answer the main question posed.

The references list needs to be reviewed in terms of the formatting. The journal title must be in italic but the paper title not and please add the DOI. Please add the other authors to all the references in the References list: use the authors names and don’t use the et al..

Considering the comments an article revision is recommended.

Comments on the Quality of English Language

English is not my mother tongue but I advise an English revision to all manuscript.

The English could be improved to express the research work more clearly and errors in words and/or sentences must be found and solved.

Reviewer 4 Report

Comments and Suggestions for Authors

The reviewed article is very interesting and well designed. The analytical methods were selected correctly. A very interesting and important issue was to find the relationship between water activity and quality characteristics of bread. I am concerned about the lack of analysis of dietary fiber with a distinction between total, soluble and insoluble fiber. The authors base their analysis on data from the oat bran supplier, but it seems to me that this is not enough in the context of the presented title, which emphasizes the role of the high fiber content in bread. Maybe it is worth rephrasing the title to emphasize the technological aspect of the work more. There is also no analysis of the content of beta glucans, which are the most important nutritional and technological components of oat bran. There is also nothing about them in the discussion. Do beta glucans play no role in shaping the structural properties of dough? I expect the discussion to be extended to include the issue of beta-glucans.
